# Resilience of *Emiliania huxleyi* to future changes in subantarctic waters

**Evelyn Armstrong** [ID]**[1,2]\***, **Cliff S. Law** [ID]**[2,3]**

**1** NIWA/University of Otago Research Centre for Oceanography, University of Otago, Dunedin, New Zealand, **2** Department of Marine Science, University of Otago, Dunedin, New Zealand, **3** NIWA, Greta Point, Wellington, New Zealand

\* evelyn.armstrong@otago.ac.nz

**Data Availability Statement:** All relevant data are available at PANGAEA, https://doi.org/10.1594/PANGAEA.959857. Armstrong, Evelyn; Law, Cliff S (2023): Physiological measurements from the long-term laboratory incubation of Emiliania huxleyi grown at present day temperature and pH

## Abstract

Lower pH and elevated temperature alter phytoplankton growth and biomass in short-term incubations, but longer-term responses and adaptation potential are less well-studied. To determine the future of the coccolithophore *Emiliania huxleyi*, a mixed genotype culture from subantarctic water was incubated for 720 days under present-day temperature and pH, and also projected future conditions by the year 2100. The future population exhibited a higher growth rate relative to present-day cells transferred to future conditions after 309 days, indicating adaptation or genotype selection; this was reflected by an increase in optimum growth temperature of ~2.5°C by the end of the experiment. Following transfer to opposing conditions in short-term cross-over incubations, cell volume responded rapidly, within eight generations, confirming trait plasticity. The changes in growth rate and cell volume were larger than reported in previous single stressor relationships and incubations, suggesting synergistic or additive effects of combined elevated temperature and lower pH and highlighting the importance of long-term multiple stressor experiments. At the end of the incubation there were no significant differences in cellular composition (particulate organic content and chlorophyll *a*), or primary production between present-day and future populations. Conversely, two independent methods showed a 50% decrease in both particulate inorganic carbon and calcification rate, consistent with the decrease in cell volume, in the future population. The observed plasticity and adaptive capacity of *E. huxleyi* indicate resilience to future conditions in subantarctic waters, although changes in cell volume and carbonate may alter grazing loss and cell ballast, so influencing carbon export to the deep ocean.

## Introduction

Ocean acidification (OA) and warming are two of the primary oceanic changes resulting from anthropogenic $CO_2$ emissions, with surface ocean pH declining by 0.1 pH units [1], and temperature rising ca. 0.6°C since 1850 [2]. Temperature and pH are among the key controls of phytoplankton growth, and consequently any change in these drivers may affect phytoplankton biomass, production and community composition [3, 4] with potential ramifications for biogeochemical budgets, carbon export and foodwebs [5]. One globally important group of

conditions in subantarctic waters and projected conditions for 2100.

**Funding:** CL and EA received funding awards from the University of Otago https://www.otago.ac.nz/ (ORF 0115-0316 Law, ORF 0117-0318 Law) and the National Institute of Water and Atmospheric Research https://niwa.co.nz (NIWA SSIF Funding in the Coasts And Oceans Centre). The funders had no role in study design, data collection and analysis, decision to publish, or preparation of the manuscript.

**Competing interests:** The authors have declared that no competing interests exist.

phytoplankton that may be susceptible to climate change are the coccolithophores [6], haptophytes with small cells ($\leq 10\mu m$ diameter), the surface of which are covered in intricate plates, coccoliths, composed of calcium carbonate. Coccolithophores comprise ca. 10% of phytoplankton biomass, and account for 50% of calcium carbonate flux in the open ocean [7]. A number of short-term studies have indicated that coccolithophores may become less calcified in the future, in response to warming and acidification, with a corresponding decrease in PIC:POC ratio [8, 9]. The resulting decrease in carbonate ballast may reduce the export of organic matter relative to primary production, the e-ratio, from the photic zone and correspondingly increase POC remineralisation at shallower depths [10]. Understanding the regional sensitivity of coccolithophores to future climate change is then central to projecting future elemental budgets and ocean-climate feedback.

The most common coccolithophore species, *Emiliania huxleyi*, is ubiquitous across much of the ocean and is extending its range into polar waters in response to warming [11, 12]. This ubiquity, and the relative ease with which it can be cultured, makes *E. huxleyi* an ideal candidate for determining climate change sensitivity. Faster growth rates and accompanying decreases in particulate organic carbon and nitrogen (POC, PON) have been recorded in *E. huxleyi* under warmer temperatures [13–15], with an inverse response at lower temperatures [14, 16]. Conversely, short-term studies of sensitivity to low pH have provided variable results, with growth rate and POC both increasing and decreasing [8, 14]. Although this inconsistency may be attributable to methodological differences [8, 17], experimental parameters [18, 19], or variability between strains [8, 17, 20], this complicates future prognosis for this important phytoplankton group.

Phenotypic plasticity in response to environmental change enables survival and growth [21]. Phytoplankton have relatively short generation times, on the order of ~1 d$^{-1}$, and so there is also considerable potential for adaptation [22–24], but most perturbation studies are too short to assess adaptation potential. As key questions regarding how phytoplankton populations will respond to climate can only be addressed by long-term incubation experiments, single-stressor studies over a few hundred generations are becoming more common [23, 25–29]. However, future changes in temperature, pH, and other climate-related drivers will not occur in isolation [30, 31]. Short-term studies have assessed the combined effect of elevated temperature and low pH on *E. huxleyi*, and shown a range of antagonistic, additive and synergistic responses [32–34]. Although, long-term studies of *E. huxleyi* adaptive capacity have predominately focused on low pH [23, 35], the few studies examining both temperature and pH have identified contrasting results for growth rate, PIC and POC content and no change in calcification or photosynthetic rate [19, 36]. Consequently, projecting future changes in *E. huxleyi* in response to long term exposure to multiple climate stressors, and the resulting biogeochemical implications is currently confounded.

Coccolithophores are particularly important in subantarctic waters, as evidenced by the "Great Calcite Belt" that circumnavigates the globe between 38◦S and 60◦S [37], with these waters also accounting for one-fifth of the global ocean carbon uptake [38–40]. However, this region has received less attention with respect to phytoplankton physiological controls, with subantarctic phytoplankton often omitted from global comparisons of growth rates [41, 42]. As coccolithophores are becoming increasingly prevalent in these waters [9, 11] it is important to determine their long-term response to projected regional pH and temperature changes. Here we present the results of a long-term (2 year) incubation of a subantarctic *E. huxleyi* strain maintained at present day (Year 2015) subantarctic mean temperature and pH, and also projected temperature and pH for the Year 2100. Growth rate was measured throughout this period, with cells periodically harvested for estimates of cell size, and additional 'cross-over' experiments were performed in which cells incubated under present-day and future conditions

were moved to the opposing conditions, to examine the plasticity of growth rate and cell volume. At day (D) 670, elemental composition and carbon and calcium uptake rates were compared across the four treatments. The study aimed to determine i) the potential for adaptation to future conditions, and ii) whether long-term exposure to future conditions results in significant changes in cell composition and morphometry. Unlike other studies in which significant temperature increments and pH decreases were used to examine the adaptation potential of monoclonal coccolithophore cultures [e.g. 28, 36], this study used a polyclonal culture of recently isolated *E. huxleyi* under environmentally relevant perturbations based on regional projections, with the primary aim of identifying the ecological impacts of climate change in subantarctic waters by the end of the century.

## Materials and methods

### Experimental organism

The *E. huxleyi* strain (P1406 E. hux #1) used in the current incubation was a B group strain (Nannotax3 [43]) with liths between 3 and 4 μm and the central area usually covered by a lath/membrane structure (S1 Fig). The strain was isolated from subantarctic waters at the end of the Munida Transect [44] east of New Zealand in June 2014. The *E. huxleyi* culture was not clonal as establishing clonal subantarctic phytoplankton cultures has not been possible in our experience. The coccolithophore was isolated in Aquil medium [45] but then maintained in a 10-fold dilution of the recommended addition of Guillards f/2 (Sigma G0154) to natural seawater supplemented with additional nutrients to nitrate 96 μM and phosphate 6 μM (f/20; [14]). In this medium, the strain continued to calcify throughout the experiment after treatment with the antibiotics penicillin, streptomycin and neomycin (Sigma P4083) at the recommended dosage to remove bacteria prior to the start of the incubation (November 2015). The removal of bacteria was checked by staining with 4',6-diamidino-2-phenylindole, DAPI [46].

### Incubation conditions

*E. huxleyi* was incubated under two conditions: present-day subantarctic water during summer (11˚C and pH 8.1), hereafter denoted 'N', and projected conditions for subantarctic water by 2100 (14˚C and pH 7.8; [47]), hereafter denoted 'F'. The pH was reduced by 0.3 units by bubbling 10% $CO_2$ through the medium prior to the addition of coccolithophore cells and confirmed using a spectrophotometric method [48].

Incubation set-up–*E. huxleyi* was grown in 250 ml acid-cleaned, sterile, polycarbonate bottles (Nalgene), with lids that were modified with 2 openings. Special air mixes (21% oxygen in nitrogen, with 380 ppm $CO_2$ for N cultures and 750 ppm $CO_2$ for F cultures; BOC Gas NZ) were passed through an inlet into the bottle headspace via a 0.22 μm syringe filter (Millipore) to maintain target pH with an exhaust with 0.22μm filter attached to relieve pressure. This set-up was preferable to bubbling the culture medium, as some phytoplankton do not tolerate this procedure [49, 50]. The cultures were maintained in two low temperature incubators (Percival LT-36VL) at 60 μmol $m^{-2}$ $s^{-1}$ in a 12 hour: 12 hour light: dark cycle.

### Culture maintenance

One culture of each treatment was maintained for the total incubation period (to D 720–725), with replicate sub-cultures prepared at the cross-over points. Every 1–2 days culture subsamples were removed after mixing and the dark-adapted *in vivo* chlorophyll *a* (chl-a) fluorescence measured with a Turner 10-AU fluorometer. The growth rate was calculated from the least squares regression of the natural logarithm of *in vivo* fluorescence versus time during

exponential growth (7–8 generations; [51]). Cultures were maintained in a semi-batch style to ensure the cells remained in exponential growth, with cell concentration maintained at low levels (<80000 cells ml$^{-1}$) to maintain pH at the target value. The pH of the medium after cell growth was checked periodically using a spectrophotometric method [48] and was always found to be at the target pH.

## Cross-over incubations

These were performed on five occasions during the two-year incubation. Triplicate cultures of N and F cells, and also the crossover experiments—F cells inoculated into N medium and incubated under N conditions (hereafter 'F in N'), and N cells incubated in F medium under F conditions (hereafter 'N in F')—were prepared as above. Crossovers have been used in previous long-term studies [36] but this is the first study to examine plasticity to combined pH and temperature changes in cross-overs. Growth rates of each culture were monitored by measuring the *in vivo* chl-a of subsamples daily, as described above. Cell size of these cultures was measured from D309, after 7–8 generations while still in exponential growth phase. Cells in 1% glutaraldehyde fixed subsamples were mounted in a nanoplankton chamber (Phytotech) and measured with a calibrated eye-piece graticule under an Olympus IX70 inverted microscope. As the fixation process caused coccoliths to separate, the cell size measurements were for the cell body only, which allows more accurate determinations of cell volume [13]. Cell volume was calculated assuming that the cells were spherical.

## Cellular composition at D670

At the final cross-over an additional suite of measurements were performed. For *in vitro* chl-a measurement, chl-a was extracted from the cells in 90% acetone and measured using a Turner 10-AU fluorometer [52, 53]. Particulate organic carbon (POC) and particulate organic nitrogen (PON) cell content was determined in cells collected on ashed GF/F (Whatman) filters with particulate inorganic carbon (PIC) removed by flooding the filter with 0.1N sulphuric acid on the filtration manifold [54]. Dried filters were packed in tin capsules before analysis using a ThermoFlash 2000 CHN Elemental Analyser. PIC was determined by measuring total carbon in a similar way to POC, and then subtracting the POC. Particulate organic phosphorus (POP) content was measured according to the methods of Solorzano and Sharp [55]. Cell counts were made at the same time as cell size was measured.

## Rate measurements at D670

Uptake rates of dissolved inorganic carbon (DIC) into POC (photosynthetic rates) were measured using C-14 labelled sodium bicarbonate (Perkin Elmer; [52, 56]). Unlabelled DIC was measured in subsamples of culture fixed with saturated $HgCl_2$ that had been stored in glass vials with rubber sealed caps until analysis using an Airica micro DIC system (Marianda, Germany). Calcium uptake rate was measured using Ca-45 (Perkin Elmer) as a tracer for total Ca uptake. Briefly, Ca-45 (0.5–1 x 10$^{-4}$ mg) was added to 10 ml culture subsamples and incubated for 6 to 7 hours in the light. Cells were then collected on GF/F filters and the radioactive label counted by scintillation counter (Perkin Elmer) after the addition of Hi-Safe II scintillant (Perkin Elmer). The total calcium uptake was calculated in a similar manner to that for inorganic carbon uptake using C-14 tracer [56]. The unlabelled calcium content in the culture medium required for this calculation was measured by inductively coupled plasma mass spectrometry. In a previous experiment measuring Ca-45 only, it was determined that 97% of the Ca-45 taken up by the cells was incorporated into coccoliths [57]. Assuming there is no fractionation within the cell then 97% of the total Ca uptake would be used for calcification.

### Temperature response curve

At the end of the incubation (D720) a temperature response experiment was performed using the N and F populations. N and F populations were grown in triplicate 28 ml Nalgene Oak Ridge-style centrifuge tubes (Sigma) in N or F medium, respectively. The tubes were held in an aluminium temperature gradient block similar to Thomas et al. [58], that was heated at one end and cooled at the other using water maintained at constant temperature by refrigerated circulators (Julabo GmbH). The temperature regime resulted in a range of 10 temperatures along the length of the block from ~ 6˚C at one end to ~ 16.5˚C at the other, with a range within replicates at each temperature of 0.1–0.2˚C. The block was lit from below which achieved a light level of 62 +/- 7 $\mu$mol m$^{-2}$ s$^{-1}$ at the bottom of the tubes in a 12 hour: 12 hour light: dark cycle. To ensure uniform light each replicate was manually rotated through the row holes at the set temperature during the incubation period. The experiment was maintained in a 4˚C controlled temperature room which enhanced the stability of the block temperatures. Measurements of *in vivo* chl-a were made daily and growth rates calculated as above. As experimental conditions in the temperature response experiment differed to the long-term incubations and cross-overs, e.g. the incubation tubes were only illuminated from the bottom, the temperature response rates were not comparable.

### Statistics

Statistical analysis was performed using Student's t-test and one-way, and one and two-way repeated measures (rm) ANOVA, as detailed in the Results section, followed by pairwise multiple comparison Student-Newman-Keuls post-hoc test using SigmaPlot v 14.5 (Systat Software, Inc.). Differences were deemed significant at the $p < 0.05$ level.

## Results

### Specific growth rate

Despite variability within both populations over the 2 year period the overall growth rate for F was faster than for N, with specific growth rates for N of 0.55–0.65 d$^{-1}$ and 0.6–0.8 d$^{-1}$ for ($F_1$ = 56.169, $p \leq 0.001$, one-way rm ANOVA, Fig 1). Linear regression indicates that growth rates of both populations increased across the incubation period, although r$^2$ values were lower for N (0.151) relative to F (0.554). Comparison of the five initial and final growth rates (D10 –D41 relative to D695 –D725 for N, D12-D54 relative to D695—D720 for F) showed a mean increase of 18% for N and 42% for F, although this was only significant for F ($F_3$ = 18.766, $p < 0.001$, one-way rm ANOVA, post hoc $F_{end}$ vs $F_{initial}$ $p < 0.001$). Furthermore, growth rates at the end of the incubation ca. D720 were significantly faster at 24.4% for F relative to N ($p$ = 0.001). The uniform growth rate in the N treatment has also been observed with Antarctic diatoms and a *Phaeocystis* species [51], which together preclude laboratory selection for higher growth rates, in contrast to other *E. huxleyi* studies [28].

### Growth rates at cross-overs

**Temporal variability in growth rate within treatments.** The specific growth rate at the five cross-over time points from D132 onwards was relatively constant for N (mean 0.63 ± 0.007 d$^{-1}$), with only the D481 rate being significantly higher ($F_{12}$ = 5.409, $p < 0.001$, two-way rm ANOVA; post-hoc D481 vs D132-670 $p \leq 0.003$, Fig 2). In contrast, the growth rate of F showed a significant increase to D481 ($p \leq 0.018$, 0.64 ± 0.02 d$^{-1}$ D132, 0.73 ± 0.004 d$^{-1}$ D481), after which it remained constant to D670. Consequently, the F growth rate was almost 20% faster than N by the end of the experiment at D670. No temporal trend was apparent in

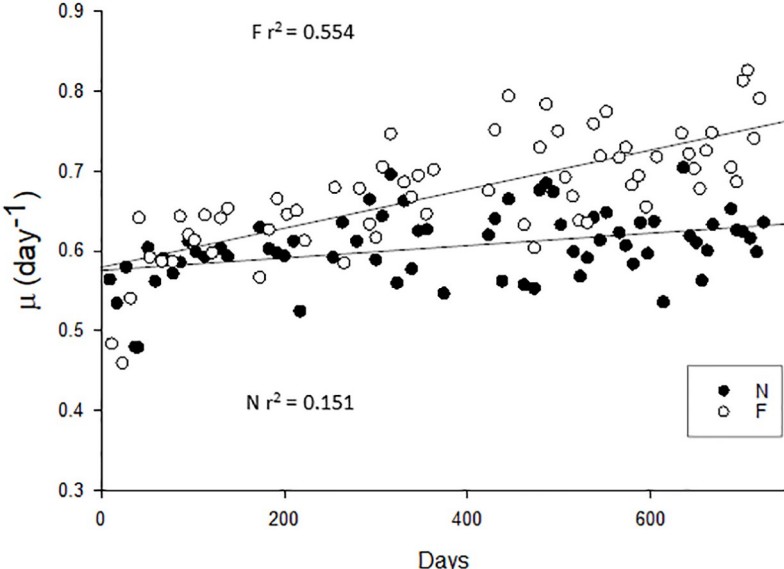

**Fig 1. Growth rate per day in N and F treatments over the 720 day incubation period.** Cultures were maintained in exponential growth with growth rates for semi-batch cultures calculated from least squares regressions of the natural log of chl-a against time. N closed circles, n = 69; F open circles n = 65. Regression for F treatment = 0.578 + (0.000244 x days) $r^2$ = 0.554. Regression for N treatment = 0.575 + (0.0000774 x days) $r^2$ = 0.151.

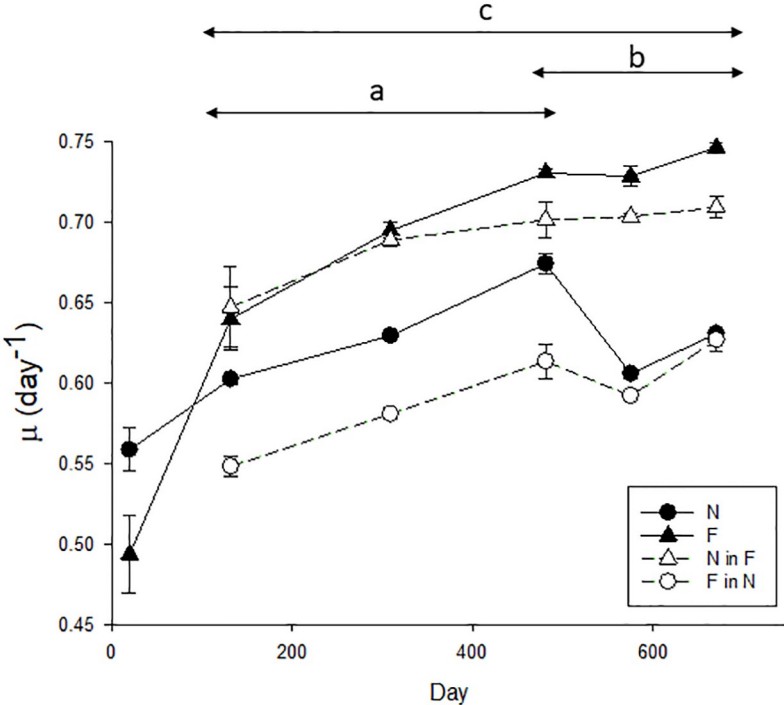

**Fig 2. Growth rate at five cross-over time points in each treatment.** Mean growth rate at each time point (n = 3), with error bars indicating standard error of the mean (SEM). The mean growth rate of the first 5 semi-continuous N and F batch cultures are included at D30 for comparison. a–growth rate of N cells is significantly greater than those in F in N, b–growth rate of F cells is significantly greater than those in N in F, c- growth rate of F cells is significantly greater than those in N.

growth rate in the N in F cross-over, except at D132 where growth was significantly lower than at all subsequent cross-over timepoints (p≤0.002). F in N growth rate increased significantly to D481 (p≤0.32) and remained at this level to D670.

**Differences in growth rate between treatments.** At all cross-over timepoints specific growth rates of F, and also N in F (warmer, lower pH) were significantly faster (p≤0.02, Fig 2) than for N, and F in N (colder, higher pH). The growth rate of N increased by 4–16% when transferred to future conditions (N in F) and matched the F growth rate for the first two timepoints. However, the F growth rate was subsequently faster than for N in F (D481 onwards; p = 0.014, p = 0.032 and p = 0.003, respectively), indicative of adaptation or genotype selection in the F population. Conversely, the F in N growth rate was lower than in F, with a decrease of 14.4–18.7%. At the first three cross-over timepoints (D132, 309 and 481) the F in N growth rate was significantly lower than for cells continuously cultured in N conditions (p<0.001). For completeness, the mean of the initial five N and F growth rate measurements at the start of the incubation (D30) are included in Fig 2 although, as there were no cross-overs at this timepoint, rates were not included in the two-way ANOVA.

## Temperature response curve

Phytoplankton species generally exhibit a consistent unimodal response across a temperature range, that is skewed above the optimum growth temperature as growth rates decline more rapidly [59]. As this study focused on projected increases to the Year 2100 the higher experimental temperature range did not extend to a full thermal reaction norm e.g. Boyd et al. [41]. Nevertheless, after 720 days the temperature response curve of the N and F populations differed (Fig 3), with significantly faster growth at mid-high temperatures (10.4–16.5°C) in the F population ($F_9$ = 6.146, p<0.001, two-way ANOVA; post hoc p≤0.049). In addition, the optimum growth temperature increased by ~2.5°C, from 13°C to 15.3°C, consistent with the experimental temperature increment between N and F.

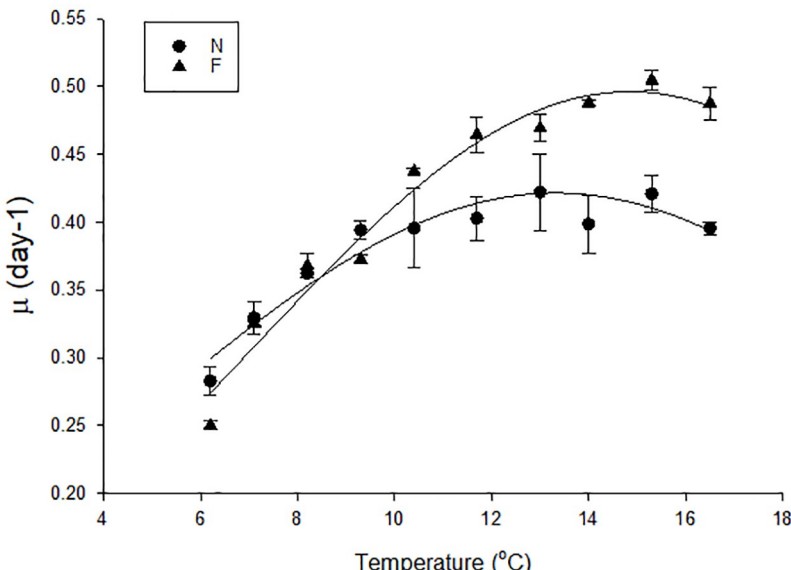

**Fig 3. Temperature response curves of N and F populations after 720 days.** Mean specific growth rate at each temperature, n = 3 except for N at 14°C, and F at 10.4°C and 15.3°C where n = 2, with error bars indicating SEM.

## Cell volume

There was a clear treatment effect on cell volume from D300, with cells incubated at present-day temperature and pH (N, and F in N) being significantly larger than those at higher temperature and lower pH (F, and N in F), as indicated in Fig 4 ($F_3$ = 27.138, p≤0.001, two-way rm ANOVA). Between D309 and D670 the cell volume range for N was 25.6–33.1 $\mu m^3$ and 16.7–21.4 $\mu m^3$ for F, with lower values of 15–40% in F relative to N at the same time point. A comparable trend was apparent for F in N (20.9–29.4 $\mu m^3$) and N in F (15.5–18.4 $\mu m^3$), with a 25–37% higher volume when F cells were transferred to N conditions, and a 36–46% decrease when N cells were transferred to F conditions. Despite the difference in the size of cells incubated in N and F conditions at different timepoints there was no significant difference.

## Cell composition

Only minor variations in cell composition were apparent between treatments on D670, with N cells containing less chl-a than F on a volumetric basis ($F_3$ = 6.024, p = 0.019 one way ANOVA, post hoc p = 0.013; S3A Fig), and POP content varying with treatment (S3C Fig). Cellular POC did not differ between treatments (Fig 5A), except on a per unit volume basis ($F_3$ = 5.628, p = 0.023 one way ANOVA) with F cells containing more POC than N (p = 0.029; S3D Fig). However, there were differences in cellular PIC ($F_3$ = 15.339, p<0.001 one way ANOVA), which was significantly greater in N and F in N, (p≤0.032), being twice the concentration (11–20 pg PIC/cell) of F, and N in F (6–10 pg/cell, Fig 5B). PIC:POC values were consistent with the broad range reported in the literature for *E. huxleyi* (0.1–2.7, [9]). Although PIC:POC was higher in N (1.58, range 1.33–1.95) relative to F (0.92, range 0.55–1.1), with a significant difference between groups ($F_3$ = 5.000, p = 0.031 one way ANOVA), there were no significant differences in the all pairwise multiple comparison (Fig 5C). However, combining the results for present-day and projected treatments (i.e. N + F in N, and F + N in F) revealed an overall significant difference (p = 0.002, Student's t-test) between these conditions.

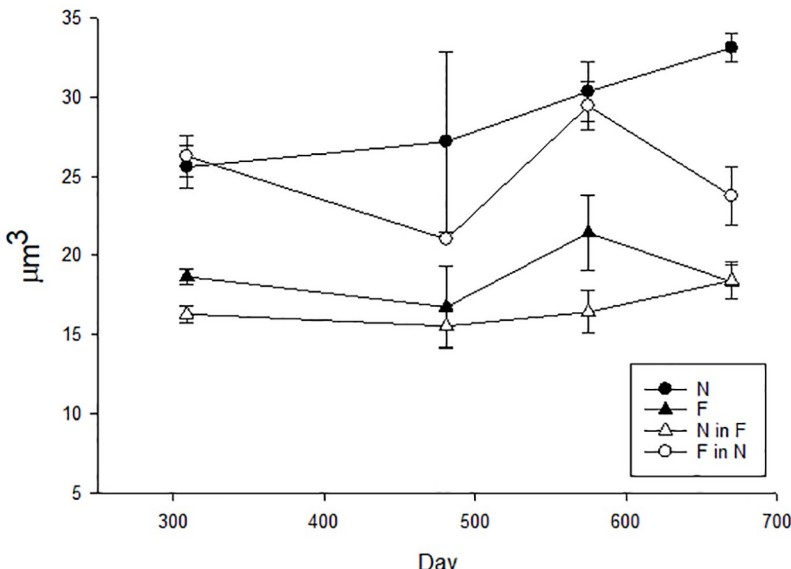

**Fig 4. Cell volumes for each treatment.** Mean volume at each time point, n = 3 except N at 481 days and F in N at 309 and 481 days where n = 2, error bars indicate SEM.

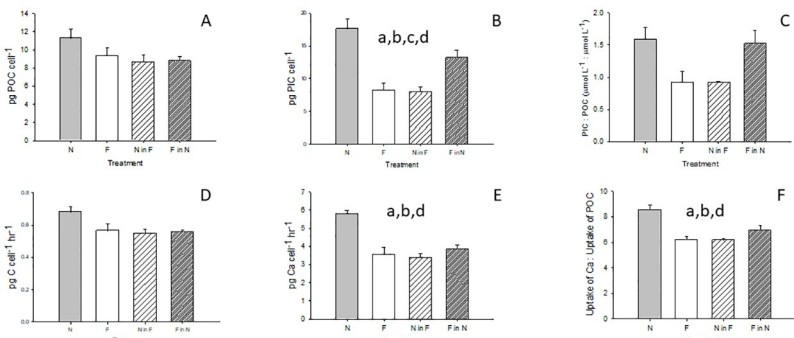

**Fig 5. *E. huxleyi* cell carbon parameters at D670 for each treatment.** Mean presented for all parameters, n = 3, error bars indicate SEM. A. POC content (pg cell$^{-1}$); B. PIC content (pg cell$^{-1}$); C. PIC:POC ratio (μmol L$^{-1}$: μmol L$^{-1}$); D. C uptake rate (pg C cell$^{-1}$ hr$^{-1}$); E. Ca uptake rate (pg Ca cell$^{-1}$ hr$^{-1}$); F. Ca uptake: C uptake (pg Ca cell$^{-1}$ hr$^{-1}$: pg C cell$^{-1}$ hr$^{-1}$). Significant differences between treatments indicated by a for N and F, b for N and N in F, c for F and F in N and d for N and F in N.

## Primary production and calcification

Incorporation rates of dissolved inorganic carbon into cellular organic carbon ranged between 0.55–0.68 pg C cell$^{-1}$ hr$^{-1}$ on D670 and reflected the observed responses in cellular POC (Fig 5A), with no significant difference between treatments (Fig 5D). Rates of calcium uptake ranged between 3.3–5.8 pg Ca cell$^{-1}$ hr$^{-1}$ (Fig 5D) and mirrored the response of cellular PIC (Fig 5B), with significantly higher Ca uptake in N, at almost twice the rate of the other treatments ($F_3$ = 16.265, p<0.001 one way ANOVA; post hoc p = 0.001). The mean calcification: photosynthesis ranged from ca. 6.2–8.5 (Fig 5F), with a significantly greater ratio in N relative to the other treatments ($F_3$ = 12.183, p<0.002 one way ANOVA; post-hoc p≤0.007).

## Discussion

Maintenance of the coccolithophore *E. huxleyi* for 720 days under future conditions projected for subantarctic water identified a significant increase in growth rate of 24% relative to present-day conditions. Increased growth rate has been reported in a combined high temperature and low pH long-term study using *E. huxleyi* [36], in contrast to long-term studies of *E. huxleyi* under lower pH alone where growth rates decreased [23, 28]. Although full factorial experiments were not carried out in this study, the future growth rate increase supports the contention that elevated temperature may counteract the detrimental effect of lower pH, as indicated in short-term experiments [33, 34, 60, 61]. Krumhardt et al. [9] predict a 10% increase in coccolithophore growth rate for a 2–3°C temperature rise, whereas Fielding's [62] power law equation predicts a 21.7% increase from 11°C to 14°C, consistent with the observed 24% increase in growth rate in F. This indicates that temperature is the dominant factor and that any negative influence of lower pH is countered by elevated temperature, so emphasising the importance of multiple stressor studies to decipher future trends.

The subantarctic *E. huxleyi* strain exhibited physiological plasticity, with the growth rate of N cells increasing within eight generations on transfer to future temperature and pH (see Fig 2). Growth rates in F and N in F were initially similar but diverged from D309 with the higher growth rates in the F population indicative of adaptation. A similar but inverse response was evident in the reverse scenario in which F cells were transferred to present day temperature and pH (F in N, see Fig 2). Adaptation to a higher temperature reduces performance at lower temperatures [36, 63], as indicated by the slower growth rate of F in N cells relative to N under

original conditions between D 132 and D590. Their subsequent convergence from D590 was due to a decline in growth rate of N, rather than a change in F in N, which cannot be readily explained. Changes in growth rate trends over time have been noted in other long-term studies of coccolithophores [27, 64]. This time-frame is consistent with the adaptation threshold of 100–400 generations reported in long-term exposure of *E. huxleyi* clonal cultures to elevated temperature and lower pH [28, 36], and also freshwater phytoplankton under elevated temperature [25, 65].

This response was not solely attributable to future conditions being more optimal for growth for the subantarctic *E. huxleyi* strain, but instead reflected adjustment of temperature tolerance to higher temperature under long-term exposure, as indicated by the increase in the growth optimum from 13˚C in the N population to 15.3˚C in the F population (Fig 3). Adjustment of optimal growth temperature to environmental temperature is well documented [42, 66, 67], and this study further confirms that this shift is unaffected by coincident decrease in pH for *E. huxleyi*. A relatively minor increase in optimal growth temperature (0.7˚C) was reported for *E. huxleyi* following incubation at +9.5˚C above ambient temperature for 2.5 years [68], in contrast to the current study where incubation at +3˚C resulted in a corresponding +2.5˚C increase in optimum growth temperature. This anomaly may reflect that the high incubation temperature in the earlier study was near the extreme for *E. huxleyi* [68], whereas the subantarctic strain in this study demonstrates the physiological capacity to adjust to smaller temperature increases that are within the range of future projections. Alternatively, the coincident decreased pH may have had an additive or synergistic effect on optimal growth rate temperature, which is consistent with the reported increase in fitness of a long-term temperature adapted population of *E. huxleyi* under decreased pH [36].

Transfer of N cells to F conditions resulted in rapid morphological adjustment within eight generations, with a significant decrease in cell volume of 27% relative to cells in present-day conditions, confirming trait plasticity. A similar response has been seen in other phytoplankton groups, with both warming [3, and references therein] and combined increased temperature and decreased pH [32, 36] reducing cell volume. Empirical observations suggest a decrease of 2.5% cell volume per 1˚C temperature rise at 15˚C [69], yet the observed cell volume change in this study was three times the predicted decrease for the +3˚C future temperature, indicating an additive, or possibly synergistic, effect of lower pH. Lower pH has been reported to both increase and decrease *E. huxleyi* cell size, depending on cell ecotype [70], pH level [23, 28, 70] and incubation duration [23, 28]. Indeed, Atkinson et al. [69] proposed that size reduction at elevated temperature may be a response to a decrease in dissolved $CO_2$, with the smaller surface to volume ratio being advantageous, as with the relationship between nutrient availability and cell size [71]. However, as neither nutrients nor $CO_2$ were limiting in this study, the alternative "compound interest" hypothesis (Lewontin, 1965 from Atkinson et al. [69]), may be more relevant, that rapid reproduction of small cells at higher temperatures is beneficial to Darwinian fitness. That the change in cell size and growth rate were inversely related suggests a trade-off in investment of energy and resources. Regardless, this combined with the rapid increase in size of F cells on transfer to N conditions, confirms the physiological plasticity of *E. huxleyi* in response to short-term variability in environmental conditions.

Despite significant changes in growth rate and cell size, there were no corresponding differences in cellular POC concentration or elemental ratios between treatments (see S2 Fig), as previously observed [19]. Cell composition remained consistent despite the decrease in cell size in the F population, resulting in higher volumetric concentrations of chl-a, POC, POP, and also higher carbon uptake rate per unit volume when compared with the N population (see S3 Fig). Significant increases and decreases in cellular POC previously reported in *E. huxleyi* long-term studies [23, 28, 36], may again reflect that larger temperature increments and

pH decreases were used relative to this study. The consistency in cellular POC suggests that the contribution of coccolithophores to subantarctic carbon budgets and foodwebs may not alter significantly in the future. Based on the changes in POC and the higher growth rate of the F population (Fig 2), this suggests that future bulk water POC will be 8% lower than for the present day. Consequently, *E. huxleyi* will continue to sustain grazer populations via increased abundance of cells of similar nutritional quality, as indicated by the cellular nutrient ratios (S2 Fig). However, the smaller cell volume may affect future trophic dynamics, with feeding size specificity potentially resulting in a shift in grazer community. The inverse relationship between phytoplankton cell size and the proportion of primary production consumed by microzooplankton [72], which are considered the dominant control of coccolithophores [73], may suggest potential increased grazing of coccolithophores which could indirectly influence the export ratio.

The two independent measures of carbonate production, cellular PIC and Ca uptake rate, in this study, showed consistent responses after 670 days, being highest in the N population and 50% lower in the F population (Fig 5). Although a decrease in PIC in *E. huxleyi* has often been reported in short and long-term incubations with lower pH alone, and in combination with elevated temperature [8, 23, 36], these results contrast with the increased PIC quota reported in a comparable study [19], supporting their contention that strain sensitivity may be a factor. The observed decreases also conflict with the reported relationship between temperature and coccolithophore PIC production [16], suggesting an antagonistic influence of elevated $CO_2$. However, the observed decrease in PIC is directly related to the cell volume response; the calculated surface area of F cells was 53% lower than for N cells, consistent with the 54% (range 34–69.5%) decrease in cellular PIC and 39% (range 20–50%) decrease in calcification rate in the F population. This is confirmed by significant linear relationships between cell surface area and PIC ($R^2 = 0.78$), and calcification rate ($R^2 = 0.79$; see S4 Fig) at D670. Inclusion of the N in F and F in N treatments further emphasises the plasticity of calcification, as is apparent in short-term studies using similar temperature and pH ranges [33, 34, 60, 74].

There is uncertainty regarding the primary function of calcification in coccolithophores, with protection from grazing, viral attack and UV radiation all considered as benefits [75–78]. Regardless, the consistency in the changes in cell volume, carbonate content and calcification (see Fig 5) infers that the benefits of calcification may remain largely unchanged in future subantarctic waters. However, the future increase in growth rate will not compensate for the decrease in total PIC, with extrapolation of the results indicating future bulk water PIC concentrations equivalent to 52% of present-day concentrations (S1 Table). The decrease in PIC: POC between N and F groups, resulting from significant decreases in cellular PIC and calcification (Fig 5) suggests future reduction in sinking particles [79] which, combined with the smaller cell volume and potential increases in microzooplankton grazing, may reduce carbon export associated with *E. huxleyi* [10]. While E. huxleyi is numerically most important in the subantarctic, other larger, less abundant coccolithophore species such as *Calcidiscus* may make a greater contribution to calcium carbonate export [80]. However, the intraspecific variation in sensitivity displayed by *E. huxleyi* illustrates the limitations of extrapolating response, and so sensitivity studies of other key coccolithophore species are also required.

Long-term adaptation will be influenced by the standing genetic capacity within the population [81], as illustrated by growth rate differences between polyclonal and single clone cultures [82], and also by different genotypes dominating at different pH levels in a long-term polyclonal *E. huxleyi* incubation [23]. Mutation can occur in both monoclonal or polyclonal cultures, whereas adaptation in multi-clonal cultures can additionally result from strain selection and sexual reproduction [22]. A mixed genotype experiment is then a more appropriate experimental strategy [20], for projecting future trends, as polyclonal cultures are more typical

of the open ocean. Regardless of the adaptation mechanism, these results extend insight into the resilience of *E. huxleyi* to climate change in subantarctic waters. Indeed, the observed acclimatory and adaptative capacity of *E. huxleyi* may contribute to its ubiquity over much of the world's ocean [83]. However, although adaptation may occur, it is not irreversible and may vary over longer time periods than used in this study, in response to genetic or energetic limitations [27, 28]. Furthermore, studies of this kind exclude other *in situ* constraints, such as nutrient and light limitation, predation and competition, that will undoubtedly be critical in determining phytoplankton success in the future ocean. Nevertheless, the observed resilience to projected warming and acidification indicates that *E. huxleyi* will continue to play an important role in subantarctic food webs and carbon cycling.

## Supporting information

**S1 Fig. Scanning electron micrographs of *Emiliania huxleyi* (P1406 E. hux #1) grown at 2015 temperature and pH.**
(DOCX)

**S2 Fig. *E. huxleyi* cell contents and ratios at D670 for each treatment.**
(DOCX)

**S3 Fig. *E. huxleyi* volumetric contents and ratios at D670 for each treatment.**
(DOCX)

**S4 Fig. Relationship between *E. huxleyi* cell surface area ($\mu m^2$) and A. PIC (pg cell$^{-1}$) and B. calcification (pg cell$^{-1}$ hr$^{-1}$).**
(DOCX)

**S1 Table. Calculation to determine decrease in future bulk water PIC content.**
(DOCX)

**S1 Appendix. Regression analyses.**
(DOCX)

## Acknowledgments

Thank you to our many colleagues at Otago, Kim Currie, Judith Murdoch, Sylvia Sander, Inda Ardiningsih, Malcolm Reid, Daryl Braid and Garth Tyrell who assisted with this study.

## Author Contributions

**Conceptualization:** Evelyn Armstrong, Cliff S. Law.

**Data curation:** Evelyn Armstrong, Cliff S. Law.

**Formal analysis:** Evelyn Armstrong, Cliff S. Law.

**Funding acquisition:** Evelyn Armstrong, Cliff S. Law.

**Investigation:** Evelyn Armstrong, Cliff S. Law.

**Methodology:** Evelyn Armstrong, Cliff S. Law.

**Project administration:** Evelyn Armstrong, Cliff S. Law.

**Resources:** Cliff S. Law.

**Writing – original draft:** Evelyn Armstrong, Cliff S. Law.

**Writing – review & editing:** Evelyn Armstrong, Cliff S. Law.

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
