## [Decision Letter · Decision Letter 0]

6 Mar 2023

PONE-D-22-32774Resilience of Emiliania huxleyi to future changes in subantarctic watersPLOS ONE

Dear Dr. Armstrong,

Thank you for submitting your manuscript to PLOS ONE. I'm sorry for the delay for the decision concerning your work. We had problems with reviewers selection and technical issues in uploading comments by rev#2. After careful consideration, we feel that it has merit but does not fully meet PLOS ONE’s publication criteria as it currently stands. Therefore, we invite you to submit a revised version of the manuscript that addresses the points raised during the review process.

I fully agree with reviewers, the paper 'Resilience of Emiliania huxleyi to future changes in subantarctic Waters' is very interesting, especially for the novel approach, but still needs an effort to be published. You find reviewers' comments appended below.==============================

We look forward to receiving your revised manuscript.

Kind regards,

Alessandro Incarbona

Academic Editor

PLOS ONE

Reviewers' comments:

Reviewer's Responses to Questions

**Comments to the Author**

1. Is the manuscript technically sound, and do the data support the conclusions?

Reviewer #1: Yes

2. Has the statistical analysis been performed appropriately and rigorously? 

Reviewer #1: Yes

3. Have the authors made all data underlying the findings in their manuscript fully available?

Reviewer #1: No

4. Is the manuscript presented in an intelligible fashion and written in standard English?

Reviewer #1: Yes

5. Review Comments to the Author

Reviewer #1: Dr Evelyn Armstrong and Dr Cliff S. Law evaluate the interactive effect of changes on multiple environmental drivers (temperature and pH) on the physiological response of the model coccolithophore species Emiliania huxleyi. Mixed genotype strains retrieved from subantarctic waters were incubated for two years under present-day and future (year 2100) conditions of pH and temperature. In order to evaluate the plasticity of E. huxleyi, authors used “cross-over” experiments in which cell cultures under present and future conditions were exposed to the opposing conditions. Based on these results, authors conclude that adaptation or a shift in the dominant strain occurred in their cultures. Moreover, the documented changes in physiological rates and cell volume for future conditions were larger than previous studies employing single drivers, which suggest that the combined effect of temperature and pH induce larger changes than variations in one single stressor alone. Overall, the results of this study indicate that E. huxleyi will adapt to future conditions in the subantarctic zone but it will reduce its cell volume and CaCO3 content with consequent impacts on the ballast efficiency and grazing.

A large body of evidence indicate that coccolithophores are sensitive to projected changes in oceanic conditions driven by ongoing human-induced climate change, such as ocean acidification, warming and changes in nutrient supply (among others). Given their abundance and fundamental role in the biological and carbonate counter pumps, changes in coccolithophore performance will most likely have impacts in the oceanic carbon cycle and marine ecosystems. Therefore, there is an urgent need of studies such as the one presented here to evaluate how multiple environmental drivers will affect key marine organisms and ecosystems.

This manuscript was a pleasure to read, it is clearly written, the results are interesting, the figures are appropriate, and the findings are useful for the scientific community. Therefore, I recommend acceptance of this manuscript after the comments listed below have been considered and implemented (if considered appropriate).

- Authors identified the culture strain as morphotype B. Are authors talking about the B group or morphotype “B”? Since the delimitation between morphotype B, B/C, C and type “O” is often difficult, it would be useful for the specialized reader to have access to a more detailed description of the morphology of the coccoliths. Moreover, some SEM pictures of the coccospheres and coccoliths would be valuable for the reader (even as a supplement).

- Müller et al. (2015) documented that B/C morphotype (which corresponds to E. huxleyi variety aurorae) is the most sensitive SO morphotype to CO2 and that could even cease coccolith production under future conditions. Did authors document any “naked” cell? Please consider this publication and include it in your discussion if considered relevant.

- E. huxleyi is the most important coccolithophore species in terms of cell abundance but not in terms of its contribution to CaCO3 (and therefore ballast). The important role of less abundant but bigger taxa such Calcidiscus leptoporus in aggregate ballast efficiency in the subantarctic zone should be acknowledge in order to put into context the results of this study. Moreover, do authors think that their observations could be expanded to other coccolithophore species dwelling in the subantarctic zone?

Reference

Müller, M.N., Trull, T.W., Hallegraeff, G.M., 2015. Differing responses of three Southern Ocean Emiliania huxleyi ecotypes to changing seawater carbonate chemistry. Marine Ecology Progress Series 531, 81-90.

Reviewer#2: The study by Armstrong and Law is an interesting and timely piece of original research. The authors add to the growing body of experimental long-term studies on coccolithophores. Their approach differs from previous ones mainly in the selection of two environmental parameters (pH and temperature) and a polyclonal, as opposed to monoclonal, culture. This is a good choice in that it better represents the situation in the ocean. The experiment is well thought through and carefully conducted. I have a few questions and suggestions that will hopefully help to improve the manuscript.

Line 111: why is a clonal culture not possible? Did you not isolate single cells?

Line 116: was the antibiotics treatment successful? Did you check for bacteria?

Line 125: Did the gas flow not lead to enhanced evaporation? Did you measure salinity?

Line 136: How was the conversion of fluorescence to cell density done? How many days / generations were included in the regression?

Line 139: high cell density usually does not lead to an increase in calcite production. At any rate, the reference is not appropriate for this statement because Hoppe et al (2011) were using cell densities up to 72,000 cells per ml only.

Line 173: calcification is strongly light dependent. Calcification might be saturated at lower light intensities than photosynthesis (Zondervan et al 2007) but it nevertheless shuts down in darkness (Kottmeier et al 2020). The argument here is concerned with the L/D cycle, i.e. total darkness. Therefore, I am not sure I understand the argument.

Line 177: your 45 label results in ca 2 permil Ca45 (my calculation assuming 10mM Ca as background; correct?). Are you saying that 97% of these 2 permil are taken up? If so, how? Is there such a strong fractionation for Ca45? Can you show the ICP-MS data and explain the calculation in more detail?

Line 330: At the end of the experiment N and F in N are the same (Fig 2). How do you explain this in terms of adaptation?

Line 338: I agree if the statement is based on Fig 3. But Fig 2 seems to tell a slightly different story (see previous point).

Line 387: cellular PIC is a quota, not a production, i.e. it does not include time. PIC production is another term for calcification rate (not recommended though, because different people mean different things, which is confusing). You measured short-term calcification rate by means of Ca45. I suggest also calculating bulk PIC production in the traditional way, i.e. PIC production = growth rate * PIC quota (e.g. Hoppe et al 2011). That would be a very interesting addition, plus it would allow direct comparison with many culture studies, because bulk PIC production is a standard parameter whereas short-term Ca45 rate is hardly ever measured.

Line 405: a normal coccolith morphology might be a central part of the benefit of calcification (Jaya et al 2016). If you do not have data on morphology this should be mentioned as a caveat here.

Line 407: how was that calculated?

Line 409: There is a distinction here between individual cell sinking rate and sinking rate of marine snow particles. The bulk PIC/POC as measured here will influence the sinking rate of marine snow in the way you describe it. But to say something about individual cell sinking rate the PIC quota of coccospheres needs to be determined by means of SEM. This is necessary because E. huxleyi cultures might contain large amounts of loose coccoliths that will contribute to bulk PIC quota but not to coccosphere PIC quota (Milner et al 2016, Rosas-Navarro et al 2018).

Refs

Hoffmann, R., Kirchlechner, C., Langer, G., Wochnik, A. S., Griesshaber, E., Schmahl, W. W., and Scheu, C.: Insight into Emiliania huxleyi coccospheres by focused ion beam sectioning, Biogeosciences, 12, 825â€“834, https://doi.org/10.5194/bg-12-825-2015, 2015.

Hoppe CJM, Langer G, Rost B. 2011. Emiliania huxleyi shows identical responses to elevated pCO2 in TA and DIC manipulations. J. Exp. Mar. Biol. Ecol. 406, 54â€“62 (doi:10.1016/j.jembe.2011.06.008)

Jaya BN, Hoffmann R, Kirchlechner C, Dehm G, Scheu C, Langer G. Coccospheres confer mechanical protection: New evidence for an old hypothesis. Acta Biomater. 2016 Sep 15;42:258-264. doi: 10.1016/j.actbio.2016.07.036. Epub 2016 Jul 21. PMID: 27449337.

Kottmeier, D.M., TerbrÃ¼ggen, A., Wolf-Gladrow, D.A. and Thoms, S. (2020), Diel variations in cell division and biomass production of Emiliania huxleyiâ€”Consequences for the calculation of physiological cell parameters. Limnol Oceanogr, 65: 1781-1800. https://doi.org/10.1002/lno.11418

Milner, S., Langer, G., Grelaud, M. and Ziveri, P. (2016), Ocean warming modulates the effects of acidification on Emiliania huxleyi calcification and sinking. Limnol. Oceanogr., 61: 1322-1336. https://doi.org/10.1002/lno.10292

Rosas-Navarro A, Langer G, Ziveri P (2018) Temperature effects on sinking velocity of different Emiliania huxleyi strains. PLoS ONE 13(3): e0194386. https://doi.org/10.1371/journal.pone.0194386

6. PLOS authors have the option to publish the peer review history of their article (what does this mean?). If published, this will include your full peer review and any attached files.

Reviewer #1: No

---

## [Author Response · Author response to Decision Letter 0]

27 Mar 2023

PONE-D-22-32774

Resilience of Emiliania huxleyi to future changes in subantarctic waters

PLOS ONE

Dear Dr Incarbona

In response to the additional requirements:

1. Regarding laboratory protocols; all of the methods used during this study are standard protocols used for phytoplankton experiments or are full described in the manuscript so probably not applicable for a separate publication. 

2. Our data has been submitted to Pangaea and is now being checked and processed. [PANGAEA-ISSUES] (PDI-34418) Data submission 2023-03-20T03:04:43Z (Evelyn Armstrong, University of Otago). Once the manuscript is accepted the moratorium on the results will be removed.

We would like to thank the reviewers for their constructive comments. Please find our responses (in bold) to the reviewers’ specific comments below. Line numbers in our responses correspond to the line numbers in the marked-up copy of the manuscript.

Reviewer 1 

- Authors identified the culture strain as morphotype B. Are authors talking about the B group or morphotype “B”? Since the delimitation between morphotype B, B/C, C and type “O” is often difficult, it would be useful for the specialized reader to have access to a more detailed description of the morphology of the coccoliths. Moreover, some SEM pictures of the coccospheres and coccoliths would be valuable for the reader (even as a supplement).

Apologies for the confusion; we are talking about B group. We have amended the text to clarify this (in line 108) by inserting ‘with liths between 3 and 4 µm and the central area usually covered by a lath/membrane structure (Fig SA)’. We have also included 3 SEM images in the supplementary material Fig SA. 

- Müller et al. (2015) documented that B/C morphotype (which corresponds to E. huxleyi variety aurorae) is the most sensitive SO morphotype to CO2 and that could even cease coccolith production under future conditions. Did authors document any “naked” cell? Please consider this publication and include it in your discussion if considered relevant.

As mentioned in lines 115 - 116 the E. huxleyi isolate continued to calcify after antibiotic treatment and we have now added ‘throughout the experiment’ at line 116 to emphasise this point. We did not see naked cells at any time during the experiment. Muller et al 2015 is cited in the manuscript at line 365. 

- E. huxleyi is the most important coccolithophore species in terms of cell abundance but not in terms of its contribution to CaCO3 (and therefore ballast). The important role of less abundant but bigger taxa such Calcidiscus leptoporus in aggregate ballast efficiency in the subantarctic zone should be acknowledge in order to put into context the results of this study. Moreover, do authors think that their observations could be expanded to other coccolithophore species dwelling in the subantarctic zone?

To address this issue we have inserted the text below at lines 432 – 436.

While E. huxleyi is numerically most important in the subantarctic, other larger, less abundant coccolithophore species such as Calcidiscus may make a greater contribution to calcium carbonate export (83). However, the intraspecific variation in sensitivity displayed by E. huxleyi illustrates the limitations of extrapolating response, and so sensitivity studies of other key coccolithophore species are also required.

83. Rigual Hernández AS, Trull TW, Nodder SD, Flores JA, Bostock H, Abrantes F, et al. Coccolithophore biodiversity controls carbonate export in the Southern Ocean. Biogeosciences. 2020;17:245-260.

Reviewer 2

Line 111: why is a clonal culture not possible? Did you not isolate single cells?

In our experience of isolating cultures from oceanic subantarctic water, establishing a culture from a single cell is not possible. This is because single cells selected from natural subantarctic phytoplankton communities and transferred to new medium do not replicate and establish cultures. In addition, as we noted (and as agreed by the reviewer), we feel it is more appropriate to use polyclonal cultures when considering the response of natural populations.

Line 116: was the antibiotics treatment successful? Did you check for bacteria?

Yes, antibiotic treatment was successful, and the culture was checked with DAPI. Now inserted at line 118: ‘The removal of bacteria was checked by staining with 4’,6-diamidino-2-phenylindole, DAPI (84).’ 

84. Porter KG, Feig YS. The use of DAPI for identifying and counting aquatic microflora. Limnology and Oceanography. 1980;25(5):943-948.

Line 125: Did the gas flow not lead to enhanced evaporation? Did you measure salinity?

Although gas flow may have enhanced evaporation, the medium was not bubbled and instead the headspace was continuously flushed. While we did not measure salinity, the gas flow rate was very low and the air movement did not disturb the surface of the culture so we expect any evaporation to be minimal. 

Line 136: How was the conversion of fluorescence to cell density done? How many days / generations were included in the regression?

Fluorescence was not converted to cell density. As mentioned at line 138 the natural logarithm of in vivo fluorescence was used for the regression. 

At line 153 the number of generations is mentioned. For clarity, this has now also been inserted at line 139 (‘7 – 8 generations’).

Line 139: high cell density usually does not lead to an increase in calcite production. At any rate, the reference is not appropriate for this statement because Hoppe et al (2011) were using cell densities up to 72,000 cells per ml only.

The reviewer is correct; the comment regarding calcite production and the reference was an error and the sentence has been removed, lines 142 – 143. 

Line 173: calcification is strongly light dependent. Calcification might be saturated at lower light intensities than photosynthesis (Zondervan et al 2007) but it nevertheless shuts down in darkness (Kottmeier et al 2020). The argument here is concerned with the L/D cycle, i.e. total darkness. Therefore, I am not sure I understand the argument.

When measuring carbon uptake and calcification using C-14, cells are incubated in both light and dark, with the dark uptake subtracted from uptake in the light to account for any attachment of the C14 label to the cells. This is usually <1 – 3 % of total C uptake. We used an alternative/ independent method of Ca-45 incorporation, during which Ca uptake occurred primarily in the light (ca. 65-75%) but also at a lower rate (ca. 25-35%) in the dark. This dark uptake did not reflect cell attachment, as cells were washed following incubation, and we assume the Ca uptake in the dark is subsequently used during calcification (though we do not infer that calcification occurs in the dark). For the calculation we did not subtract dark uptake from uptake in the light, consistent with the published method of Satoh et al ( ref 56). 

As we do not present results for Ca uptake in the dark or light, and further studies are required to investigate this, we have rephrased the comparison of C14 and Ca uptake in the Methods section (as below), and removed mention of dark uptake, to address the reviewer’s comments. 

Line 177: your 45 label results in ca 2 permil Ca45 (my calculation assuming 10mM Ca as background; correct?). Are you saying that 97% of these 2 permil are taken up? If so, how? Is there such a strong fractionation for Ca45? Can you show the ICP-MS data and explain the calculation in more detail?

Too clarify, we assume that 97% of the Ca45 is incorporated in to coccoliths, as identified by Satoh et al (ref 56). We do not assume fractionation for Ca45, or that cells preferentially take up Ca45 over unlabelled Ca. 

Lines 175 – 183 have been replaced with lines 183 - 194 and clarified as follows:

“Calcium uptake rate was measured using Ca-45 (Perkin Elmer) as a tracer for total Ca uptake. Briefly, Ca-45 (0.5 – 1 x 10-4 mg) was added to 10 ml culture subsamples and incubated for 6 to 7 hours in the light. Cells were then collected on GF/F filters and the radioactive label counted by scintillation counter (Perkin Elmer) after the addition of Hi-Safe II scintillant (Perkin Elmer). The total calcium uptake was calculated in a similar manner to that for inorganic carbon uptake using C-14 tracer (54). The unlabelled calcium content in the culture medium required for this calculation was measured by inductively coupled plasma mass spectrometry. In a previous experiment measuring Ca-45 only, it was determined that 97% of the Ca-45 taken up by the cells was incorporated into coccoliths (56). Assuming there is no fractionation within the cell then 97% of the total Ca uptake would be used for calcification. “

Line 330: At the end of the experiment N and F in N are the same (Fig 2). How do you explain this in terms of adaptation?

We have edited the text to reflect the reviewer’s observation at line 346:

“Adaptation to a higher temperature reduces performance at lower temperatures (36, 63), as indicated by the slower growth rate of F in N cells relative to N under original conditions between D 132 and D590. Their subsequent convergence from D590 was due to a decline in growth rate of N, rather than a change in F in N, which cannot be readily explained. Changes in growth rate trends over time have been noted in other long-term studies of coccolithophores (27, 71).”

Line 338: I agree if the statement is based on Fig 3. But Fig 2 seems to tell a slightly different story (see previous point).

Fig 2 (Line 346 ) and Fig 3 (Line 358) are not directly comparable. In Fig 2 the F in N treatment is maintained long term at elevated temperature AND CO2 and transferred to short-term exposure at the original temperature AND CO2, whereas in Figure 3 both treatments are only exposed to changes in temperature. Furthermore, we have explained in the response above that the convergence between F&N and N after D590 is due to an unexplained decrease in N growth rate. 

Line 387: cellular PIC is a quota, not a production, i.e. it does not include time. PIC production is another term for calcification rate (not recommended though, because different people mean different things, which is confusing). 

We appreciate that PIC/cell is not a rate, although it does provide an indication of PIC production. To reduce confusion we have edited ‘calcification rate’ at line 404 to ‘Ca uptake rate’.

You measured short-term calcification rate by means of Ca45. I suggest also calculating bulk PIC production in the traditional way, i.e. PIC production = growth rate * PIC quota (e.g. Hoppe et al 2011). That would be a very interesting addition, plus it would allow direct comparison with many culture studies, because bulk PIC production is a standard parameter whereas short-term Ca45 rate is hardly ever measured. Would there be assumptions in recasting this as a rate?

We have calculated PIC production (see figure below) as suggested by the reviewers and achieve the same statistical differences between the four treatments as presented in Fig 5E Ca uptake rate. We measured Ca-45 uptake as a direct and independent indicator of calcification rate, rather than use particulate inorganic carbon content, which is determined indirectly from the difference between total particulate and particulate organic carbon, and then applied to growth rate. As our data will be freely available on Pangaea, it will be possible for anyone interested in direct comparison to calculate these values. 

FIGURE INCLUDED HERE IN RESPONSE TO REVIEWER FILE.

Line 405: a normal coccolith morphology might be a central part of the benefit of calcification (Jaya et al 2016). If you do not have data on morphology this should be mentioned as a caveat here.

While we find the reviewer’s comment unclear and wonder if the morphology data is related to the previous comment about the cells being naked as the Jaya et al 2016 paper discusses compressing coccolithophores. As mentioned previously our cells were not naked. However, we have added the Jaya et al reference at line 423 as it adds to the literature regarding the function of liths. 

Line 407: how was that calculated?

The following text and table has been added to the supplementary data to explain the calculation:

Table S1. Calculation to determine decrease in future bulk water PIC content. From application of one days growth at the N growth rate at D670 (0.631) one cell = 1.88 cells with a PIC of 33.1 pg. At the F growth rate at D670 (0.746) one cell = 2.11 cells after one days growth, with a PIC content of 17.39 ng. The PIC content after one days growth at 2100 is only 52% of the PIC content at 2015. 

CORRECTLY FORMATTED TABLE BELOW CAN BE SEEN IN "REPONSE TO REVIEWER FILE'

 Growth rate at D670 PIC Content at D670 (ng per cell) Cell number after one day of growth Total PIC content of cells after growth (ng) %age of N PIC content at F conditions

N 0.631 17.6 1.88 33.09 

F 0.746 8.24 2.11 17.39 52.5%

Line 409: There is a distinction here between individual cell sinking rate and sinking rate of marine snow particles. The bulk PIC/POC as measured here will influence the sinking rate of marine snow in the way you describe it. But to say something about individual cell sinking rate the PIC quota of coccospheres needs to be determined by means of SEM. This is necessary because E. huxleyi cultures might contain large amounts of loose coccoliths that will contribute to bulk PIC quota but not to coccosphere PIC quota (Milner et al 2016, Rosas-Navarro et al 2018).

We did not discuss sinking rate in our manuscript, only the bulk content of particles that may sink. To avoid confusion, we have edited this to read ‘future reduction in sinking particles’ at line 430.

Finally, we have edited ref 48 to Shi et al 2009 as Iglesias-Rodriguez et al was incorrect and amended the access date for ref 82 to the last date. 

Shi D, Xu Y, Morel FMM. Effects of the pH/pCO2 control method on medium chemistry and phytoplankton growth. Biogeosciences. 2009;6:1199–1207.

We hope you find these edits and comments to be satisfactory.

Yours sincerely

Evelyn Armstrong

---

## [Editor Report · Decision Letter 1]

30 Mar 2023

Resilience of Emiliania huxleyi to future changes in subantarctic waters

PONE-D-22-32774R1

Dear Dr.Armstrong,

We’re pleased to inform you that your manuscript has been judged scientifically suitable for publication and will be formally accepted for publication once it meets all outstanding technical requirements.

Kind regards,

Alessandro Incarbona

Academic Editor

PLOS ONE

---

## [Editor Report · Acceptance letter]

5 Jul 2023

PONE-D-22-32774R1 

Resilience of *Emiliania huxleyi* to future changes in subantarctic waters 

Dear Dr. Armstrong:

I'm pleased to inform you that your manuscript has been deemed suitable for publication in PLOS ONE. Congratulations! Your manuscript is now with our production department. 

Kind regards, 

on behalf of

Professor Alessandro Incarbona 

Academic Editor

PLOS ONE